# Worry perception and its association with work conditions among healthcare workers during the first wave of the COVID-19 pandemic: a web-based multimethod survey at a university hospital in Sweden

Eirini Alexiou [1,2] Helle Wijk,[3,4,5] Magnus Åkerström [6,7] Ingibjörg H Jonsdottir [6,7] Alessio Degl' Innocenti,[2,8] Linda Ahlstrom [4,9]

For numbered affiliations see end of article.

**Correspondence to**
Dr Eirini Alexiou;
eirini.alexiou@vgregion.se

## ABSTRACT

**Objectives** In this study, we explored healthcare workers' (HCWs) worry perception and its association with their work situation during the first wave of the COVID-19 pandemic.

**Design** A web-based multimethods survey including multiple choice and open-ended questions was used.

**Setting** The study was conducted at a university hospital in Sweden.

**Participants** All HCWs who were working during the first wave of the COVID-19 pandemic in March–June 2020 were eligible. HCWs (n=6484, response rate=41%) from 69 departments fulfilled the study inclusion criteria and responded to the survey. Of them, we analysed data from the 3532 participants who replied to the open-ended questions (54% of the respondents).

**Main outcomes measures** Worry perception and its association with work conditions among HCWs.

**Results** 29% (n=1822) and 35% (n=2235) of the responding HCWs experienced a daily or more than daily strong worry of being infected or infecting others with SARS-CoV-2. This finding could be further confirmed and explored with themes from the qualitative results: 'ambiguity of feeling safe and secure', 'being obliged to adapt to a new reality' and 'into the unknown'. The themes consisted of 6 main categories and 15 subcategories. The findings revealed that the two main drivers of worry perceived by HCWs were lack of personal protective equipment and fear of bringing the virus home to their families and friends.

**Conclusions** Worries of getting infected are common among HCWs during crises such as the COVID-19 pandemic. Several factors are raised that plausibly could minimise the negative effects of worry among HCWs. Thus, effective preventive work plans should be created, promoted and communicated in order to minimise the effects of such crises and support HCWs. By focusing on effective communication and preparedness, including access to relevant protective equipment and providing general support to HCWs, the work environment and patient care could be sustained during a crisis such as the COVID-19 pandemic.

## STRENGTHS AND LIMITATIONS OF THIS STUDY

⇒ The key strength of this study lies in its transdisciplinary research cooperation, with several multidisciplinary researchers actively participating in the implementation of the study. Additionally, the analysis and interpretation of the data and findings were bolstered by the use of the NVivo software program. This enhanced the quality of the analysis, making the data more manageable, ordered and facilitating the creation of categories for easy comparison and discussion. Ultimately, this approach aided in reaching a consensus on the categories.

⇒ Data collection during the pandemic was conducted during a relatively calm period between the first and second waves, which started around November 2020 in Sweden.

⇒ One study limitation is that the quantitative data were cross-sectional, preventing the drawing of conclusions about cause and effect relationships between variables.

⇒ The single-hospital setting and the relatively low response rate limit the generalisability of the findings.

⇒ Another limitation was that, due to pandemic restrictions, the research methodology exclusively used written answers and no face-to-face interviews were conducted; this approach presented challenges in facilitating a dynamic interaction with the participants as the researchers could not ask follow-up questions to further develop and explore the answers.

## INTRODUCTION

The COVID-19 pandemic meant a tremendous increase in the workload for healthcare workers (HCWs), thereby exacerbating pressure on an already strained workforce and causing additional distress.[1] Worry refers to the thoughts, images, emotions and actions of a negative nature in a repetitive,

uncontrollable manner that result from a proactive cognitive risk analysis made to avoid or solve anticipated potential threats and their potential consequences.[2] In other words, worry is a natural response to anticipated future problems.[3] However, when the threat is uncertain and continuous, or when worry is not well calibrated with the actual threat, such as as during the COVID-19 pandemic, worry can become excessive and burdensome.[4]

The COVID-19 outbreak in China in December 2019 and in Europe in February 2020 caused widespread worry, which when at healthy levels led to adaptive, protective behavioural changes. However, for some individuals, the pandemic outbreak led to excessive, maladaptive levels of worry. The experience of worries has been shown to be mainly related to perceived risks for loved ones, possible health problems related to COVID-19 and perceived social restrictions during the pandemic.[5] High levels of worry during the COVID-19 pandemic have been indicated in several surveys.[6 7] In a cross-sectional study conducted in Thailand, 100% of the included HCWs reported worry and fear of COVID-19 related to infection control practices.[8] Another cross-sectional study found that HCWs reported high levels of worry about getting infected and developing COVID-19, which indicates the great extent to which HCWs were highly worried about such scenarios.[9] In a cross-sectional study involving HCWs, a high prevalence of depression, anxiety and stress (77%–57%) was observed; workplace worries, including concerns about personal infection, transmission to others, inadequate personal protective equipment (PPE) and insufficient training in PPE usage, were identified as contributing factors.[10] This underlines the need to address the likely mental health and linked retention crisis in the healthcare workforce and provide support to create a psychologically safe working environment.

Both front-line and non-front-line HCWs are at risk of SARS-CoV-2 infection and may therefore perceive significant worries. Early on during the pandemic, COVID-19-associated mental health distress became a notable problem among front-line HCWs[11 12] and a great number of HCWs became infected.[13 14] Front-line HCWs who took care of COVID-19 patients daily were at higher risk of developing worries compared with HCWs who were not involved in the treatment of patients.[15] A systematic review reported that increased levels of potential exposure during the COVID-19 pandemic, as well as in previous coronavirus outbreaks, were shown to cause long-term worries.[16] Despite this, the importance of providing support in terms of guidelines, policy documents and sufficient resources to HCWs during and after the pandemic to ensure the future of healthcare staff has been neglected.[1]

In a previous study, we investigated the effects of the COVID-19 pandemic on the working environment of HCWs compared with the prepandemic situation.[17] The results showed that worries about getting infected were among the several factors that were associated with HCWs' perceptions of their work situation. Thus, the rationale of this study was to extend this knowledge by exploring HCWs' perceptions about worry and its association with perceived working conditions, such as workload and emotional support, during the first wave of the COVID-19 pandemic in Sweden. Understanding the factors related to HCWs' worries could be used to design, promote and communicate preventive measures to help alleviate the adverse impacts of such worries and support HCWs in similar future situations, thereby improving HCWs' working conditions and counteracting the risk of severe distress and burnout.

## METHODS

This study is part of a larger research project investigating how the work environment and health of HCWs were affected by the COVID-19 pandemic. The current study used a multimethod design comprising both qualitative and quantitative data from a survey containing multiple choice and open-ended answers. Thus, the survey data with fixed questions were used to analyse self-reported quantitative data regarding worry about being infected and infecting others. The fixed survey questions were also used to analyse which factors were related to the reported level of worry using the Strengthening the Reporting of Observational Studies in Epidemiology cross-sectional reporting guidelines.[18] The open-ended questions were used to analyse participants' perceptions related to worry about being infected and infecting others.

### Setting

The study was conducted at a large university hospital in northern Europe, situated in West Sweden. The hospital provides emergency and basic care for more than 1 million inhabitants in the region and offers highly specialised care for the 1.7 million inhabitants of West Sweden. During the pandemic, the hospital provided intensive care for patients with symptomatic COVID-19.

### Population and procedure

A web-based COVID-19 survey was administered to all hospital HCWs (n=17 914) regardless of having contact with patients (COVID-19 or in general) or having non-clinical work tasks.[18] After excluding employees who were absent from work during the study period (n=1399), 16 515 were eligible for study participation, of whom 6816 responded to the survey, resulting in a response rate of 41% (figure 1).

During the first week of September 2020, an invitation to participate was sent by email that includes a link to an anonymous survey. One reminder was provided during the last week of September 2020. The participants were given approximately 5 weeks to answer the survey. When answering the questionnaire, study participants were asked to keep in mind how they perceived their working conditions during

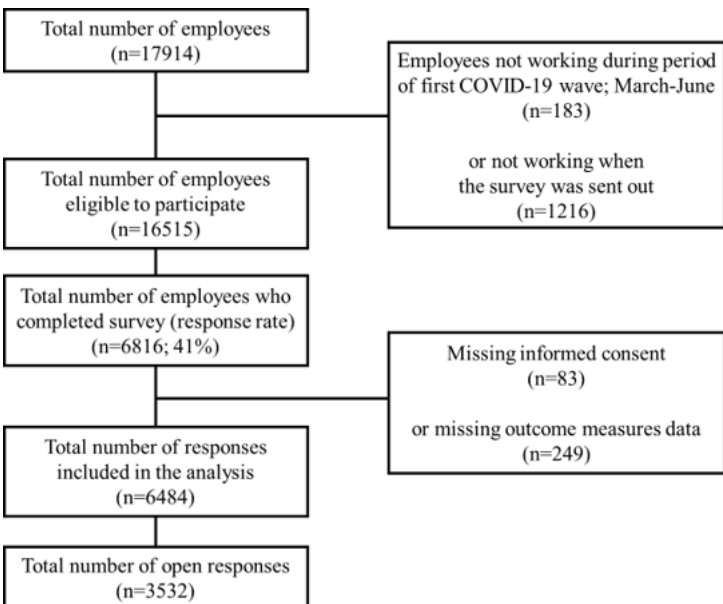

**Figure 1** Participant flow chart.

the intensive period of the pandemic in the spring of 2020.

### Ethical considerations
Participants provided informed consent.

### Survey and outcome measures
The survey was designed to be completed in 10–20 min. Demographic information, such as age, gender, organisational affiliation, professional role, specialist training, working hours (daytime, evening, night shifts or varied hours) and eleven individual items related to work conditions, addressing work demands, support, recovery and engagement, were collected.[17] Additional items about work placement during the pandemic, worries about getting infected and access to PPE were included. Participants were additionally asked to reflect on an open-ended question about their positive and negative experiences during the first wave of the pandemic. The multimethod research involved the use of two complementary data sets and analysing approaches in parallel, guided by a shared overarching research aim.[19]

### Qualitative analysis
In the qualitative part, designed to capture new insights and knowledge regarding the phenomenon of worry among HCWs during the first wave of COVID-19, we used a qualitative inductive content analysis of the responses to the open-ended question, performed in three phases: (1) preparation phase, (2) organising phase and (3) reporting phase.[20 21] In the first phase, the first and last authors carefully read the text separately to obtain their own comprehensive understanding. Thereafter, they each selected a suitable unit of analysis and data that align with the intended focus in order to enhance credibility, and thereafter condensed the data. In the second phase, data were coded and grouped, and subcategories,

main categories and themes were created and discussed until consensus by EA and LA and further abstracted with HW. Data were compared for similarities, differences and patterns to enhance the confirmability of the findings. The data analysis included an iterative process, with regular reconciliations and discussions, with the larger research group. Working strategically prevented researcher bias and ensured data coding was externally heterogeneous and internally homogenous. A computer-assisted qualitative data analysis software (NVivo V.12; QSR International) was used to aid data organisation. Trustworthiness was ensured by following the steps for data collection and reporting of the results, as outlined by Elo *et al*.[21] Analysis also followed the Consolidated criteria for Reporting Qualitative research, a 32-item checklist for interviews.[22]

### Quantitative analysis
The fixed survey questions were used to analyse the prevalence of perceived worries among the participants and which factors affected the level of worry among the respondents. Individuals without informed consent (n=83), missing data on all work environment items (n=21) or missing an organisational affiliation (n=211) were not included in the analysis. Excluded individuals were evenly distributed among departments and professional roles. One administrative department had limited respondents (n=7) and was excluded due to the risk of identification of individuals. The percentage of responding HCWs with a strong worry about being infected and infecting others during the first wave of the COVID-19 pandemic was summarised descriptively for all survey respondents.

Mixed-effects models (Proc Mixed in SAS V.9.4; SAS Institute, USA) were used to assess the impact of age, experience of lack of PPE, being redeployed to another department, caring for COVID-19 patients and perceived

working conditions on HCWs' worry of being infected or infecting others, with these measures used as fixed effects (nested within departments) and departments as a random effect. Hypothesis testing for fixed and random effects was performed using Wald tests and likelihood ratio tests, respectively. Statistical significance was set at p<0.05 and two-sided CIs were used.

### Patient and public involvement
None.

### RESULTS
There were 6484 HCWs from 69 departments at the hospital who responded to the survey and fulfilled the inclusion criteria for this study. Of them, 3532 (54%) replied to the open-ended question, which generated about 216 pages of A4 text.

Most of the respondents were women (83%, 5348), and 30% (n=1924) were employed as registered nurses, 21% (n=1327) as assistant nurses, 10% (n=10) as physicians, 12% (n=803) as administrative staff, and the remaining (27%, n=1502) belonged to other occupations, such as midwives, psychologists, managers, technicians and physiotherapists. The respondents had the following age distribution: 12% (n=746) of participants were ≤29 years old, 22% (n=1418) aged 30–39 years, 23% (n=1482) aged 40–49 years, 27% (n=1766) aged 50–59 years and 16% (n=1045) were ≥60 years.

### Percentage of responding HCWs with a strong worry about being infected and infecting others during the first wave of the COVID-19 pandemic
A total of 28% (n=1822) and 35% (n=2235) of the responding HCWs experienced a daily or more than daily strong worry about being infected or infecting others, respectively (figure 2). There was a high correlation between expressing a strong worry about being infected and about infecting others (r=0.63, p<0.001, n=6357).

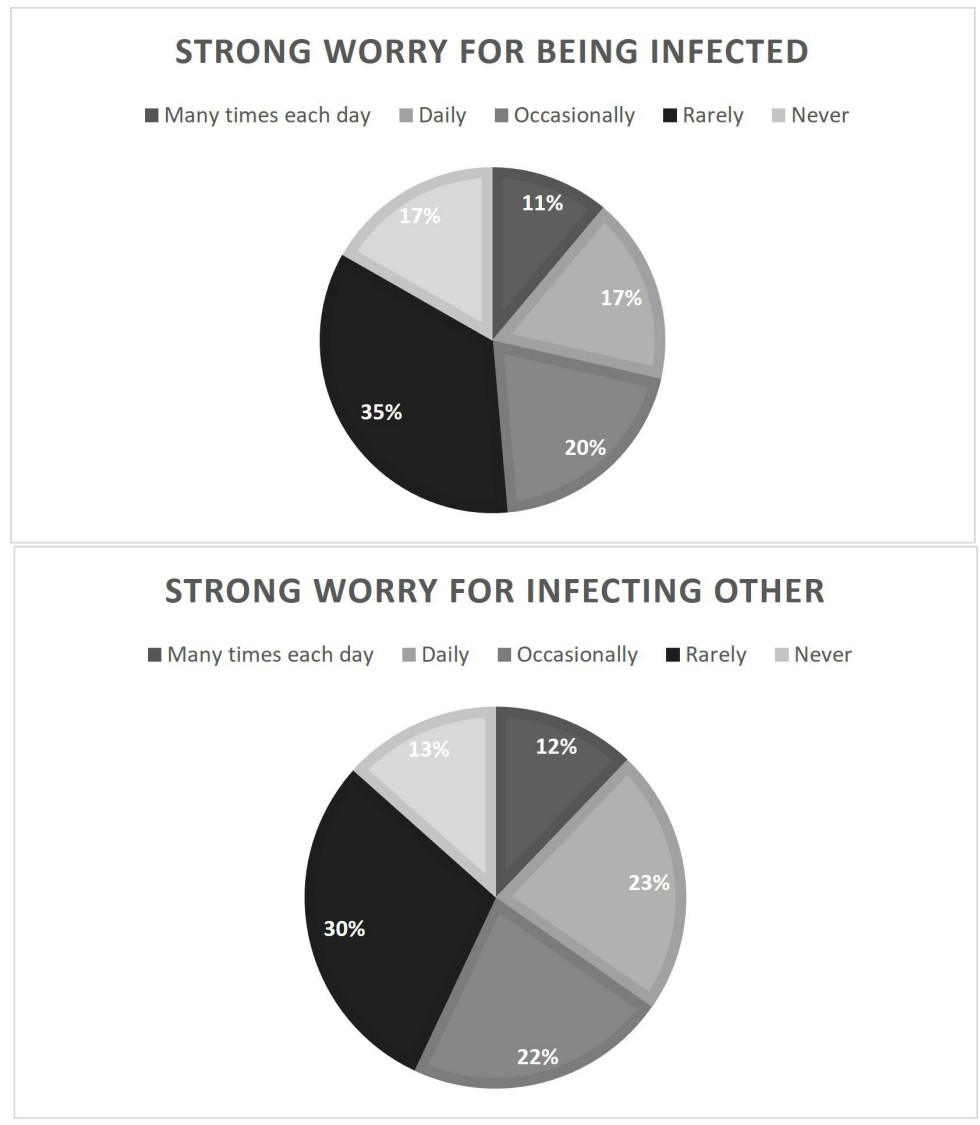

**Figure 2** Percentage of participating healthcare workers who reported a strong worry of being infected or of infecting others during the first wave of the COVID-19 pandemic.

**Table 1** Subcategories, main categories and themes concerning worry perceptions among healthcare workers during the first wave of the COVID-19 pandemic onset in spring 2020

| Subcategories | Main categories | Themes |
|---|---|---|
| Availability of personal protective equipment and testing for COVID-19 | Accessibility | Ambiguity of feeling safe and secure |
| Priorities over testing and using protective equipment | | |
| Training in proper use of personal protective equipment | | |
| Information | Communication | |
| Routines | | |
| Decisions | | |
| Learning | Competence | Being obliged to adapt to a new reality |
| Workshifting | | |
| Care priorities | Mission | |
| Patient outcomes | | |
| Infection exposure | Challenges | Into the unknown |
| What tomorrow brings | | |
| Socialising with family and friends | Social interaction | |
| Patient interaction | | |
| Interaction with colleagues | | |

## HCWs' worries during the first wave of the COVID-19 pandemic: qualitative analysis

The analysis of the qualitative data resulted in three themes: 'ambiguity of feeling safe and secure', 'being obliged to adapt to a new reality' and 'into the unknown'. The themes consisted of a total of 6 main categories and 15 subcategories (table 1).

### Theme 1: ambiguity of feeling safe and secure

This theme contains conceptions relating to the perceived concern about protection against the virus when exposed. This includes having or lacking access to PPE against COVID-19, such as face masks and/or respirators, face shields/goggles and protective clothing, as well as conceptions about being able to use the available PPE and being able to be tested for SARS-CoV-2 when experiencing symptoms. The theme consists of two main categories, 'accessibility' and 'communication', which are which are both related to HCWs' perception of their ability to protect themselves and patients from contracting the virus as either good or poor.

The main category, accessibility, covers three subcategories: availability of PPE and testing for SARS-CoV-2 infection, priorities over testing and using PPE, and training in the proper use of PPE. HCWs described feeling forced to prioritise PPE use when the availability of PPE was limited. In many cases, HCWs lacked access to PPE when needed. Furthermore, they sometimes lacked the opportunity to train and become confident in the proper use of PPE, including when to use it. The results of this study demonstrate that HCWs' lack of access to PPE and testing was not only due to material shortage but also due to prioritisation decisions and to different time schedules and priorities between departments. In general, based on HCWs' experiences, those not working on the front line were not always permitted to use PPE or undergo testing for SARS-CoV-2 infection.

> The positive has been that there is protective equipment to use and that all patients are tested regularly. —P33, assistant nurse, geriatrics department

> The negative was that I experienced a few re-evaluations of the need for protective equipment when it started to run out. It was in the middle of that period when we had the most to do and no one knew how it would end. To then claim that COVID-19 was not contagious if you had a visor (which happened to be the protective equipment we had in stock and could be reused) felt disrespectful and completely illogical. —P66, medical resident, geriatrics department and emergency department

The second main category, communication, contains conceptions about how clarity in information as well as constantly changing routines and decisions impacted HCWs' feelings of being able to protect themselves and others from the virus.

> Very unclear sometimes regarding protective equipment, especially at the beginning of the pandemic when routines regarding protective equipment were changed very often, created an uncertainty. —P2, registered nurse, paediatrics department

> Lack of clarity in routines when it came to the use of protective equipment in different situations. —P42, registered nurse, radiology department

### Theme 2: being obliged to adapt to a new reality

The second theme contains perceptions relating to feelings of inadequacy, despite doing one's best, and feelings of sadness when being incapable of providing sufficiently

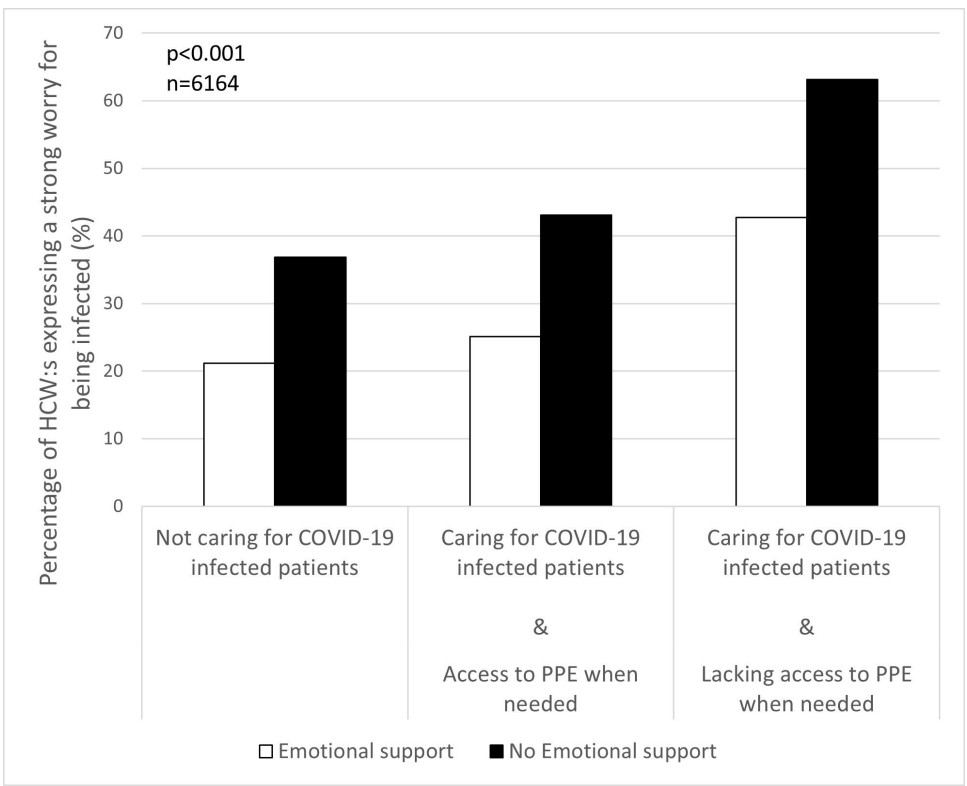

**Figure 3** Percentage of participating healthcare workers (HCWs) who reported experiencing a strong worry of being infected, stratified by caring/not caring for COVID-19 patients, access/lacking access to personal protective equipment (PPE) when needed and access/lacking access to emotional support.

good care. These feelings also addressed handling the challenges in patient care that came with the COVID-19 pandemic, for example, to have to prioritise daily patient care and make difficult decisions, while feeling scared and sad about being unable to deliver good quality care to patients.

This theme consists of two main categories: 'competence' and 'mission'. The first main category, competence, covers two subcategories: 'learning' and 'workshifting'. These subcategories are related to the feeling of having relevant clinical knowledge (or not) and the skills needed to perform new tasks at the new workplace to which HCWs were assigned during the pandemic.

Being forced to do a work for which you lack training and experience in a completely new workplace without receiving training and without a work description. —P85, registered nurse, intensive care unit

Workshifting, which we fought for a long time to happen, now happened quickly and smoothly. —P25, healthcare administrative secretary, medical secretary, emergency department

The subcategory competence was also related to the feeling of having had the opportunity to learn the skills needed to fight against the virus but also to adapt to the demands of digital transformation and to adopt new technologies by attending courses and other educational and training programmes. HCWs report mixed perceptions of adapting to the new skills and tasks, leading in some cases to increased worries and concern.

What amazing people who work in health care! To adapt quickly to new guidelines, new care, new workplace, colleagues, etc. —P146, assistant nurse, hand surgery department

The second main category, mission, includes two subcategories: 'care priorities' and 'patient outcomes'. These were mostly related to the outcomes of one's own personal efforts and the feeling of being able to protect patients from getting infection and care for patients to save their lives, as well as making priorities and having enough time to spend with the patients in need.

I felt inadequate towards the patients. I could not always provide as good care as desired due to high workload. —P24, registered nurse, infectious diseases department

The most negative thing was all the patients who died without having someone with them. Neither relatives nor HCWs. It feels hard to have been a part of that. Overall, what was experienced as difficult is that the patients with COVID (or suspected COVID) got stuck, their care was not always good. —P31, registered nurse, infectious diseases department

### Theme 3: into the unknown

The third theme contains conceptions relating to the worries about the COVID-19 pandemic and potential outcomes. It consists of two main categories: 'challenges' and 'social interaction'. Challenges covers two subcategories: 'infection exposure' and 'what tomorrow brings'. These subcategories are related to worries about being exposed to COVID-19, of oneself becoming infected, of transmitting the virus to someone else or of patients infecting each other. The second subcategory describes an emotional state caused by feared, unpredictable consequences on personal somatic health. No one knew who would be infected and the outcome of infection for any particular individual.

> I was surprised how scared I was, I'm not a scared person, but this really scared me, and it didn't feel very safe when some of my colleagues were infected at the departments where they worked. It felt good that the hospital locked its doors and that there were guards. —P21, administrative assistant, health and rehabilitation department

The second main category, social interaction, covers three subcategories: 'socialising with family and friends', 'interaction with colleagues' and 'patient interaction'. Reported worries were related to the emotional state due to feared consequences regarding those with whom one was socialising or interacting, thereby affecting the health of loved ones, colleagues and patients not yet infected.

> Negative: Strong concern that my relatives would get COVID-19. —P104, assistant nurse, intensive care unit

### Factors affecting the level of worry among responding HCWs

The worry of being infected or infecting others decreased with age (p=0.002 (n=6349) and p<0.001 (n=6418), respectively) and increased with lack of PPE (p<0.001 (n=3933) and p<0.001 (n=3975), respectively), being redeployed to another department (p=0.03 (n=5909) and p=0.05 (n=5969), respectively) and when caring for COVID-19 patients (p<0.001 (n=6354) and p<0.001 (n=6420), respectively).

To further disentangle the effect of the factors identified in the qualitative analyses, the associations between worry about being infected and caring for COVID-19 patients in combination with access to PPE when needed and perceived working conditions (ie, access to emotional support and a reasonable amount of work) were investigated. The results show that caring for COVID-19 patients, especially in combination with a perceived lack of PPE, was associated with a higher percentage of HCWs reporting a daily strong worry about being infected (p<0.001, n=6164, figure 3; p<0.001, n=6254, figure 4). Simultaneously, a lack of emotional support (figure 3) and a reasonable amount of work (figure 4) led to higher levels of strong worry.

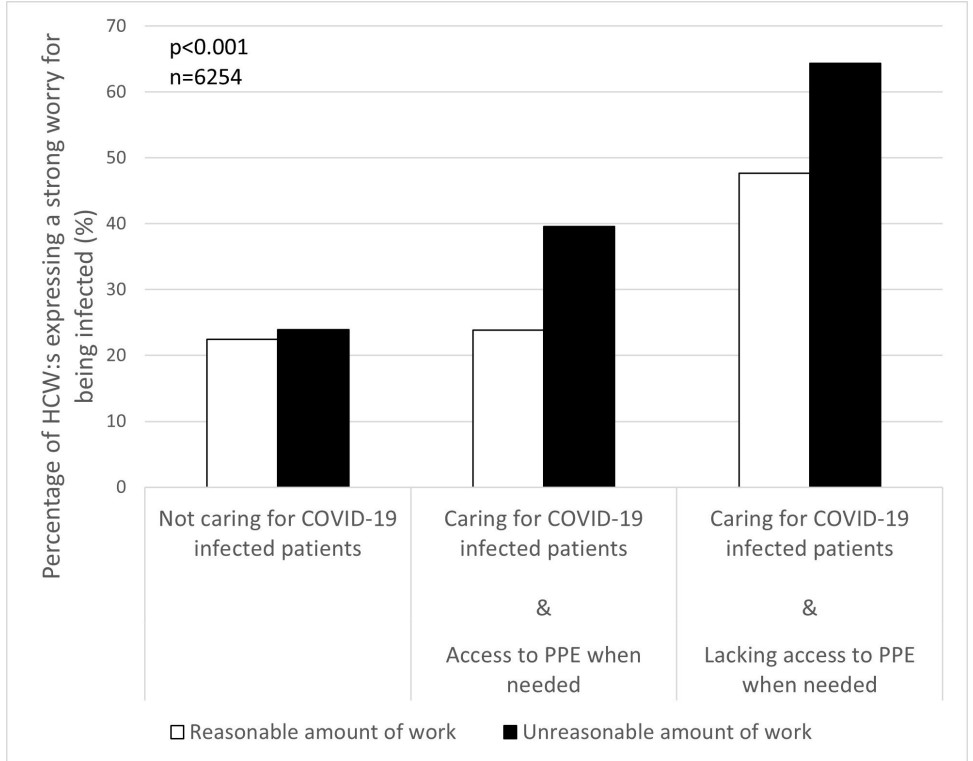

**Figure 4** Percentage of participating healthcare workers (HCWs) who reported experiencing a strong worry about being infected, stratified for caring/not caring for COVID-19 patients, access/lacking access to personal protective equipment (PPE) when needed and having a reasonable/unreasonable amount of work.

## DISCUSSION
### Main findings

The quantitative results showed that four out of five responding HCWs reported high levels of worry, half of them daily, which could be further confirmed and explored with themes from the qualitative results. These worries decreased among participants who were supported by accessible PPE and clear communication about routines and how to protect oneself. Worries related to a lack of feeling safe due to lack of PPE while working are in line with the results from previous studies.[23] The rationale for being worried was related to the rapid spread of COVID-19, its strong contagion factor and the lethality in severe cases, without any specific medication for treatment.[24]

A review study of the psychological trauma faced by HCWs in the intensive care unit during the COVID-19 pandemic reveals primary concerns centred around worries of transmitting the virus to their families, personal health worries, witnessing colleagues contracting the disease and facing stigmatisation from their communities of being contagious[25]; the same results could be seen in several previous studies.[26 27] These findings are affirmed by confirming our main category, Social Interaction, which illustrates that apprehension primarily revolves around potential consequences for those individuals with whom one is socialising or interacting. This concern is of great significance.

Nevertheless, this is surprising considering the knowledge within the healthcare sector with regard to strategies to educate and comfort HCWs about infection control. In the Swedish healthcare setting, there is a long tradition of a strong focus on adherence to hygiene routines and knowledge about preventing the spread of infection.[28–30] However, there is evidence that repeated training on the proper use of PPE,[31] together with regular updates on COVID-19 pandemic,[32] is successful in reducing worry among HCWs. Therefore, if the hospital had offered specific and intense training on infection prevention, regular COVID-19 testing and contact tracing to HCWs at an early stage, this might have decreased HCWs' worries regarding their safety and strengthened their impression of the management supporting and empowering employees by taking such actions. On the other hand, a survey from Thailand found that feelings of worry resulted in higher compliance to PPE and infection prevention practices,[8] as well as patient safety risk behaviour, thereby improving patient outcomes. Emphasising the need for managers to proactively seek information from recent evidence, identify situations causing concerns among HCWs[26] and determine the most effective ways to support and enhance HCWs' safety. This proactive approach aims to alleviate and validate the concerns of HCWs, promoting a positive workplace environment that addresses and mitigates worries.[10]

Another recommended strategy is for managers to focus on the early signs of worries among individual HCWs, offer support from psychological services,[9] work closely with their employees, and express empathy, gratitude and prudence for HCWs' valuable efforts,[32] and confirm their worries when the situation is beyond one's control.[33]

The increased burden on HCWs during crises needs to be addressed by relocating internal and external flow capacity and by training and strengthening the available workforce.[31] The skills and competencies that new crises may demand differ, and it is therefore complicated and even impossible to know what may be needed when going into the unknown. Nevertheless, there are ways for organisations to prepare. To adjust to a new reality, there is a need to ensure adequate staffing levels with the right competence and clinical knowledge skills, as well as appropriate workload and confidence to succeed with new tasks without compromising safety, quality of care and patient outcomes, even though the individual employee might have to work at a different workplace or undergo workshifting. The amount of time spent on different work tasks varies, and the schedule does not take into consideration extra time for donning PPE, additional rest days and maintaining education and training, including training new colleagues. As reported here, HCWs adapted and learnt new skills promptly and were proud and impressed by that. For this to work, organisations need to be able to better provide both emergency and planned care during and after crises, and balancing structures that strengthen organisational resilience are needed.[34] Resilience at the individual level has limited impact unless the work to strengthen the workplace also infiltrates the team and organisational levels. Rangachari and Woods[34] therefore highlight the importance of viewing organisational resilience as a system developed by managers to learn from the work of individual employees with a focus on proactivity, adaptation and recovery.

The higher level of worry among HCWs motivated them to adjust to a new, unpredictable reality. One way to address this is through the implementation of effective risk communication programmes to increase awareness of the risks associated with the pandemic and promote the adoption of appropriate preventive measures. Healthcare organisations and policymakers need to prioritise the implementation of effective risk communication strategies to promote safety among HCWs. However, there are ways to be prepared for crises in order to enhance the work environment, including improving safety. It is essential to accept that acknowledging the unknown is not a failure, and feeling worried is a natural response.[24]

Developing strategies for managing worries can further support building resilience in the face of uncertainty, hinder the consequences of professional worries on work quality and reduce the negative impact on patient outcomes.[35] In line with a multicentre study,[4] our study identified that the most frequently reported concern was for others to be infected by SARS-CoV-2 and worry about the health of loved ones. Notably, respondents in the multicentre study expressed various worries, including the impact of COVID-19 on the healthcare system, the economy, society, work loss and changes in daily routines. Our study supports these findings and demonstrates that

worries in all areas were referred to as factors that have an impact on the individual, organisational and societal levels.

## Methodological considerations

An essential step in the qualitative data analysis process involved categorising the results. This process went beyond merely grouping data that appeared similar or connected. Rather, it involved a deliberate effort to gain a deeper understanding of the phenomena under investigation by classifying the data into meaningful categories.[20] In this study, the authors' preunderstandings of the studied phenomena differed; the first, second and last authors are registered nurses who participated in the first part of the analysis. The variability in experience can be explained by the professions' respective scopes of practice and competencies in relation to their worries.

Quality criteria and recommendations in the form of checklists have been synthesised by multiple publications for researchers. However, concerns have been raised that checklists may suppress the diversity and multiplicity of practices within the qualitative paradigm and thus researchers must be aware and open-minded. As an alternative, throughout this study, best practice guidelines were applied to achieve and assess methodological rigour and research quality. Guidelines by Elo and Kyngäs[20] and the Standards for Reporting Qualitative Research checklist[22] were followed to uphold quality. We used data saturation to describe the achievement of a sufficient sample size, and ethical conduct was maintained throughout the research process. The methodology was acknowledged to be limited by researcher bias, which was minimised through predetermined plans. Multiple sources of data were used to optimise accuracy in data collection and analysis processes, resulting in credible and confirmable results.[36]

One study limitation is the relatively low response rate (41%, of which 54% replied to the open-ended questions), which affects the generalisability of the findings. Low response rates are a common problem in research in general, and in this study the response rate may also reflect the strained working conditions of HCWs during the COVID-19 pandemic. The survey was also sent out to all employees regardless of whether they were working directly with COVID-19 patients or not. This may partly explain the low response rate if employees refrained from responding as a result of feeling that the survey was not aimed towards them. In line with this, we have earlier noticed a somewhat higher response rate among frontline workers compared with, for example, administrative staff.[17]

Another issue that should be raised as a limitation is the generalisability of the results. The study was conducted in a single hospital in Sweden and it is crucial to keep this in mind when interpreting the data. However, we have no reason to believe that this hospital differs significantly from others in terms of the challenges faced by hospital staff during COVID-19. The perception of concerns among hospital staff appears to be consistent across hospitals in the western part of the world.

## CONCLUSIONS

Overall, this study raises several factors that plausibly could help in minimising the negative impact of the COVID-19 pandemic on HCWs' worries. Developing strategies for managing worries during crises should be created and promoted in order to support and strengthen HCWs. By focusing on effective communication, preparedness, including access to relevant protective equipment, and general support to the HCWs, the work environment and patient care could be sustained during crises such as the COVID-19 pandemic.

**Author affiliations**
[1]Department of Forensic Psychiatry, Sahlgrenska University Hospital, Gothenburg, Sweden
[2]Center for Ethics, Law, and Mental Health (CELAM), Sahlgrenska Academy, University of Gothenburg, Gothenburg, Sweden
[3]Department of Quality Strategies, Sahlgrenska University Hospital, Gothenburg, Sweden
[4]Institute of Health and Care Sciences, Sahlgrenska Academy, University of Gothenburg, Gothenburg, Sweden
[5]Department of Architecture and Civil Engineering, Chalmers University of Technology, Gothenburg, Sweden
[6]Institute of Stress Medicine, Gothenburg, Sweden
[7]School of Public Health and Community Medicine, Institute of Medicine, Sahlgrenska Academy, University of Gothenburg, Gothenburg, Sweden
[8]Regionhälsan, Gothenburg, Sweden
[9]Department of Orthopaedics, Sahlgrenska University Hospital, Gothenburg, Sweden

**Correction notice** This article has been corrected since it was published. Licence updated to CC BY on 1st August 2024.

**Acknowledgements** We would like to express our sincere thanks to the employees of the university hospital for their time responding to the survey. We would also like to thank Flora Cassiano and Sandra Pettersson for their valuable help with the administration of the survey and with the data analysis. Lastly, we would like to thank the human resources department at the university hospital, particularly Camilla Nilsson and Berit Roos Holmquist, for their collaboration and support in the development of the survey.

**Contributors** EA and LA designed the study, collected the data, analysed the qualitative data and drafted and revised the manuscript. HW designed the study, collected the data and drafted and revised the manuscript. MA designed the study, collected the data, made the quantitative statistical analyses and revised the manuscript. IHJ and ADI designed the study, collected the data and revised the manuscript. All authors contributed to the article and approved the submitted version. LA acts as a guarantor.

**Funding** The authors have not declared a specific grant for this research from any funding agency in the public, commercial or not-for-profit sectors.

**Competing interests** None declared.

**Patient and public involvement** Patients and/or the public were not involved in the design, or conduct, or reporting, or dissemination plans of this research.

**Patient consent for publication** Not required.

**Ethics approval** The studies involving human participants were reviewed and approved by the Swedish Ethical Review Authority (ref 2020-04771). The study was conducted in compliance with the Helsinki Declaration and the General Data Protection Regulation (EU) 2016/679. The patients/participants provided their written informed consent to participate in this study.

**Provenance and peer review** Not commissioned; externally peer reviewed.

**Data availability statement** Data are available upon reasonable request. The raw data supporting the conclusions of this article will be made available on reasonable request to the guarantor without undue reservation.

**ORCID iDs**
Eirini Alexiou http://orcid.org/0000-0001-5745-1994
Magnus Åkerström http://orcid.org/0000-0002-8469-6193
Ingibjörg H Jonsdottir http://orcid.org/0000-0003-2289-9646
Linda Ahlstrom http://orcid.org/0000-0002-3372-8722

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
