## [Reviewer comments · BMJ Open]

ARTICLE DETAILS

TITLE (PROVISIONAL)	Worry perception and its association with work conditions among healthcare workers during the first wave of the COVID-19 pandemic: a web-based multi-methods survey at a university hospital in Sweden
AUTHORS	Alexiou, Eirini ; Wijk, Helle; Åkerström, Magnus; Jonsdottir, Ingibjörg; Degl' Innocenti, Alessio; Ahlstrom, Linda

VERSION 1 – REVIEW

REVIEWER	Al-Yateem, Nabeel University of Sharjah
REVIEW RETURNED	09-Nov-2023

GENERAL COMMENTS	Thank you for inviting me to review the manuscript. I have carefully considered its content and relevance, and I offer the following feedback for improvement: The manuscript's significance is not clearly articulated in the introduction. As a reader, I was unable to grasp the study's importance. There is a pressing need for the authors to underscore the relevance and contribution of this work to the existing body of literature. The methods section lacks clarity. It is currently difficult to comprehend the study's design and execution. Additionally, the fact that the research was conducted in a single, albeit large, hospital raises concerns about the generalizability of the results. The authors must justify the rationale for this study's design, especially considering the abundance of existing literature on the topic with similar findings. More specific comments are as follows: a. The title and abstract should reflect the study's methodology more accurately. Currently, the se indicate a web survey design However later in the mthods section a multi methods design is described. b. The abstract mentions that 3,532 individuals responded to an open-ended question, yet it is unclear whether all these responses were informative or analyzed. This requires clarification. c. The strengths and limitations section lacks detail and does not effectively inform the reader. This section could be greatly enhanced to better contextualize the study's findings.
--

	d. The introduction fails to present a compelling rationale for the study. While it mentions the general anxiety provoked by COVID-19 among healthcare workers—a well-documented fact—it does not explain what new insights this study offers. e. The methodological reporting is vague. The manuscript alternates between describing a survey study and a multimethod study. A clear and consistent account of the methods is essential. f. The study's findings are limited to one hospital setting, which significantly limits their applicability to broader contexts. These issues are critical from my perspective and must be addressed to ensure the study's meaningful aspects can be reviewed and understood comprehensively. I hope these comments assist in strengthening the manuscript and I look forward to the revised version. Regards,
--	--

REVIEWER	Blanchard, Janice The George Washington University, Emergency Medicine
REVIEW RETURNED	07-Dec-2023

GENERAL COMMENTS	 1. In the introduction, I would describe how worry is different from other outcomes that have been studied during the pandemic. Most studies have focused on anxiety and depression. How is this different. At this point the study is somewhat dated. Why is this relevant now? 2. This seems to have a large sample size, there are a lot of women in your sample. This makes me wonder why this occurred and what is it about your sampling method that led to this. Is this representative of the population of healthcare workers from which you sampled? 3. It is surprising your results did not mention moral injury, a common theme that came up during other research. It is worth discussing why your results were so difference. 4. Your references seem somewhat dated, please updated as many studies have come out since 2020.
--

VERSION 1 – AUTHOR RESPONSE

Reviewer: 1	
The manuscript's significance is not clearly articulated in the introduction. As a reader, I was unable to grasp the study's importance. There is a pressing need for the authors to underscore the relevance and contribution of this work to the existing body of literature.	The manuscript's significance is now clearly articulated in the introduction illuminated by additional references.
The methods section lacks clarity. It is currently difficult to comprehend the study's design and execution. Additionally, the fact that the research was conducted in a single, albeit large, hospital raises concerns about the generalizability of the results. The authors must justify the rationale for this	We have rewritten the method section and hopefully it is clearer now. We

study's design, especially considering the abundance of existing literature on the topic with similar findings.	have also added a sentence regarding the limitation that the study was conducted in single hospital.
a. The title and abstract should reflect the study's methodology more accurately. Currently, the se indicate a web survey design However later in the methods section a multi methods design is described.	Thank you for your comment. Both the title and the abstract are now revised so they reflect the study's methodology more accurately.
b. The abstract mentions that 3,532 individuals responded to an open-ended question, yet it is unclear whether all these responses were informative or analyzed. This requires clarification.	Thank you for pointing this out. Se have now clarified that in the abstract.
c. The strengths and limitations section lacks detail and does not effectively inform the reader. This section could be greatly enhanced to better contextualize the study's findings.	We have focused on being more detailed and informative in this section.
d. The introduction fails to present a compelling rationale for the study. While it mentions the general anxiety provoked by COVID-19 among healthcare workers—a well-documented fact—it does not explain what new insights this study offers.	The rational and possible new insights to be generated is now clarified.
e. The methodological reporting is vague. The manuscript alternates between describing a survey study and a multimethod study. A clear and consistent account of the methods is essential.	We have expanded on the multimethod reporting and added a reference. Further, we are now being more detailed about the analysing process.
f. The study's findings are limited to one hospital setting, which significantly limits their applicability to broader contexts. These issues are critical from my perspective and must be addressed to ensure the study's meaningful aspects can be reviewed and understood comprehensively. I hope these comments assist in strengthening the manuscript and I look forward to the revised version.	Thank very much for this comment. We have now addressed this issue in the methodological discussion.
Reviewer: 2	
1. In the introduction, I would describe how worry is different from other outcomes that have been studied during the pandemic. Most studies have focused on anxiety and depression. How is this different. At this point the study is somewhat dated. Why is this relevant now?	You are right that many studies have focused on anxiety and depression. In our study however,

	we didn't used anxiety and depression validated scales because we aimed to explore factors that influenced HCWs' worry perception in a more exploratory manner. Understanding this could be used to design preventive measures in similar future situations and thereby improve HCWs' working conditions and prevent worry. Furthermore, defending a dated approach involves acknowledging its contributions to the scientific knowledge base.
2. This seems to have a large sample size, there are a lot of women in your sample. This makes me wonder why this occurred and what is it about your sampling method that led to this. Is this representative of the population of healthcare workers from which you sampled?	Yes, you are right. The fact that there are a lot of women in our study sample is representative of the population of healthcare workers in Sweden.
3. It is surprising your results did not mention moral injury, a common theme that came up during other research. It is worth discussing why your results were so different.	You are right that moral injury is an important issue to study and indeed relevant to study among healthcare workers during the covid-19 pandemic. However, this study aimed to explore which factors influenced HCWs' worry perception and the association of

	worries with perceived working conditions, such as workload and emotional support. Thus, studying moral injury was not a part of this study and perhaps not an issue that would turn up when studying worries.
4. Your references seem somewhat dated, please updated as many studies have come out since 2020.	Thank you for this. We have more updated references now, having made literature searches in PubMed and Scopus.
Editor(s)' Comments to the Authors:	
*We note that the response rate for the overall survey was limited (41%) and that only 54% of those that did participate contributed data in response to the open-ended questions that are the focus of the present analysis. Both the limited initial response rate and the limited proportion contributing data for analysis limit the representativeness of the sample and introduce potential bias. This should be highlighted as a key limitation and fully discussed, as well as ensuring the response rate(s) are reported in the abstract and main text. In addition, where quantitative claims are made, they should be carefully phrased to avoid over-claiming in view of these limitations (eg, "Four out of five HCWs reported high levels of worry" would be better as "Four out of five of the HCWs surveyed reported high levels of worry").	Thank you for these important suggestions. We have made a number of revisions in the manuscript which we hope to increase the transparency and clearness regarding the limited response rates. The limited response rates have been discussed more thoroughly in the strengths and limitation section page 9, line 398-406. One study limitation is the relatively low response rate (41% of which 54% replied to the open-ended question) which affect the generalisability of

	the findings. Low response rates are a common problem in research in general and in this study the response rate may also reflect the strained working conditions for HCW's under the COVID-19 pandemic. The survey was also sent out to all employees regardless of whether they were working directly with COVID-19 patients or not. This may partly explain the low response rate if employees refrained from responding as a result of feeling that the survey was not aimed towards them. In line with this we have earlier noticed a somewhat higher response rate among front-line workers compared with e.g. administrative staff (Jonsdottir et al 2021). The response rates have been included in the abstract as well as in the main text on page 3, line 129-130 and page 4, 3 line 183. To avoid over-claiming the quantitative claims, "HCW" has been
--	--

	changes to “responding HCW” throughout the manuscript.
*Related to the above point, we note a lack of numerical data throughout the abstract and main manuscript. This sometimes makes it difficult to ascertain which analysis populations are being referred to. For example, for the quantitative results, it is not clear if you included all survey respondents as the denominator or just those who also contributed to the open-ended questions. This should be clarified. Additionally, we require absolute numbers to be reported along with all percentages (in the abstract and main text). It would also be useful to include numerical data in tables where appropriate (eg, where you state “Most of the responders were women (83%)”, etc, you could then refer to a table containing the absolute numbers as well as the percentages).	Thank you for noticing. Numerical data have been added throughout the abstract, method and results section where quantitative results have been reported. In addition, on page 4, line 172, a clarification has been made that quantitative analyses were made on all survey respondents.
*Along with your revised manuscript, please include a copy of the CROSS checklist for the reporting of survey studies indicating the page/line numbers of your manuscript where the relevant information can be found (https://www.equator-network.org/reporting-guidelines/a-consensus-based-checklist-for-reporting-of-survey-studies-cross/), updating the manuscript as needed to ensure all reporting requirements are met.	This is now completed and attached
*Additionally, as part of your study is based on qualitative analysis, it might also be useful to complete a copy of the SRQR checklist for reporting of qualitative research, indicating the page/line numbers of your manuscript where the relevant information can be found (http://journals.lww.com/academicmedicine/fulltext/2014/09000/Standards_for_Reporting_Qualitative_Research___A.21.aspx), ensuring any applicable reporting requirements are met.	This is now completed and attached
*Please revise the title of your manuscript to include the research question, study design and setting. This is the preferred format of the journal. Eg, “Worry perception and its association with work conditions among healthcare workers during the first wave of the COVID-19 pandemic: a web-based mixed-methods survey of workers at a university hospital in Sweden” (or similar).	Thank you for your comment. The title is now revised according to your suggestion.
*Please revise the abstract to ensure that it is formatted according to our Instructions for Authors (http://bmjopen.bmj.com/pages/authors/#research), including all relevant subheadings and required details.	The abstract is now formatted according to the journal’s guidelines. We have included all relevant subheadings.

*Please ensure that you have highlighted the key methodological limitations of the study in the 'Strengths and limitations of this study' section and that you have fully discussed these and other relevant limitations in the Discussion section of the main text.	We have now expanded the relevant limitations of the study in the corresponding section.
*BMJ Open requires authors of all submissions to the journal to include a "Patient and public involvement" statement in the manuscript. "Patient and public involvement" should be included as a subheading at the end of the main text Methods section of all manuscripts. The statement should provide a brief description of any patient involvement in study design or conduct of the study, as well as any plans to disseminate the results to study participants. If patients and or public were not involved, please state this (eg, "None."). See our Instructions for Authors for further details: https://bmjopen.bmj.com/pages/authors/#reporting_patient_and_public_involvement_in_research	The Patient and Public statement is now included as a subheading at the end of the main text in the Methods section.
*Please complete a thorough proofread of the text and correct any spelling and grammar errors that you identify.	The manuscript has been proofread by a native English speaker and translator.
*Please note that we use Vancouver style referencing (ie, numbered references), not Harvard style in-text referencing. Please update accordingly.	The references are now updated according to the Vancouver style.
*Please move the 'Data Sharing Statement' to before the reference list (alongside the other statements), change the heading to 'Data availability statement', and update the statement to clarify how data will be shared (eg, "The raw data supporting the conclusions of this article will be made available on reasonable request to the corresponding author", or similar).	We have now moved the 'Data Sharing Statement' to before the reference list, changed the heading to 'Data availability statement', and updated the statement to clarify how data will be shared.

To ensure the system doesn't lock you out, please submit your revised manuscript at least one calendar day before 11-Jan-2024. If you are locked out and unable to submit your revision, please email our Editorial Office (info.bmjopen@bmj.com) and they will reopen it for you.

Thank you for submitting your article to BMJ Open; we look forward to receiving your revision.

If you have any queries, please contact the Editorial Office at info.bmjopen@bmj.com.